# A new method for identifying the acute respiratory distress syndrome disease based on noninvasive physiological parameters

**Pengcheng Yang[1], Taihu Wu[1], Ming Yu[1], Feng Chen[1], Chunchen Wang[2], Jing Yuan[1], Jiameng Xu[1], Guang Zhang[1] ***

**1** Institute of Medical Support, Academy of Military Sciences, Tianjin, China, **2** Department of Aerospace Medicine, Air Force Military Medical University, Xi'an, China

* zhangguang01@hotmail.com

**Data Availability Statement:** All relevant data are within the paper and its Supporting Information files.

## Abstract

Early diagnosis and prevention play a crucial role in the treatment of patients with ARDS. The definition of ARDS requires an arterial blood gas to define the ratio of partial pressure of arterial oxygen to fraction of inspired oxygen ($PaO_2/FiO_2$ ratio). However, many patients with ARDS do not have a blood gas measured, which may result in under-diagnosis of the condition. Using data from MIMIC-III Database, we propose an algorithm based on patient non-invasive physiological parameters to estimate P/F levels to aid in the diagnosis of ARDS disease. The machine learning algorithm was combined with the filter feature selection method to study the correlation of various noninvasive parameters from patients to identify the ARDS disease. Cross-validation techniques are used to verify the performance of algorithms for different feature subsets. XGBoost using the optimal feature subset had the best performance of ARDS identification with the sensitivity of 84.03%, the specificity of 87.75% and the AUC of 0.9128. For the four machine learning algorithms, reducing a certain number of features, AUC can still above 0.8. Compared to Rice Linear Model, this method has the advantages of high reliability and continually monitoring the development of patients with ARDS.

## 1. Introduction

Acute respiratory distress syndrome is a disease that seriously threatens the health of human lives[1,2]. According to relevant epidemiological investigations, the in-hospital mortality rate of ARDS is as high as 40%[3,4]. Currently, the diagnosis of ARDS disease is mainly based on the Berlin definition[5]. The Berlin definition was introduced in 2012 and allowed a clear diagnosis of ARDS disease by stating that when positive end-expiratory pressure (PEEP) $\geq 5$ cmH2O, ARDS can be classified into three stated with increasing severity, namely, mild ($200 <$ arterial oxygen partial pressure ($PaO_2$)/ fraction of inspired oxygen ($FiO_2$) (P/F) $\leq 300$), moderate ($100 <$ P/F $\leq 200$), and severe (P/F $\leq 100$), according to the level of oxygenation index (P/F). At present, blood gas analysis is mainly used to measure $PaO_2$ to calculate

**Funding:** This study was supported by the National Key Research and Development Program of China (Grant Number: 2017YFC0806402) and Tianjin Science and Technology Program (Grant Number: 18ZXJMTG00060). The work was also funded in part by logistics scientific research foundation program at the Military Medical Innovation Project (Grant Number: 16CXZ034). The National Key Research and Development Program of China (Grant Number:2017YFC0806406).

**Competing interests:** The authors have declared that no competing interests exist.

**Abbreviations:** ARDS, acute respiratory distress syndrome; ACC, accuracy; ADASYN, the Adaptive Synthetic; AUC, areas under the ROC curves; BER, balanced error rate; BMI, body mass index; CCU, Coronary care unit; CSRU, Cardiac Surgery Recovery Unit; $FiO_2$, fraction of inspired oxygen; GCS, Glasgow Coma Scale; HIPAA, the Health Insurance Portability and Accountability Act; ICU, Intensive Care Unit; k-NN, k-nearest neighbor; MICU, Medical Intensive Care Unit; MIFS, Mutual information based feature selected; MIMIC-III, Medical Information Mart for Intensive Care III; NPV, Negative predictive value; OI, Oxygenation Index; OSI, Oxygenation Saturation Index; P/F, $PaO_2/FiO_2$; $PaO_2$, partial pressure of arterial oxygen; PEEP, positive end-expiratory pressure; PHI, protected health information; PPV, Positive predictive value; ROC, receiver operating characteristic; S/F, $SpO_2/FiO_2$; SEN, sensitivity; SICU, Surgical intensive care unit; SLP–FNN, single hidden layer feedforward neural network; SMOTE, the Synthetic Minority Oversampling Technique; SPN, specificity; $SpO_2$, pulse oximetric saturation; TSICU, Trauma Surgical Intensive Care Unit.

the P/F value to evaluate the severity of ARDS. However, this method is still limited by some defects[6]. Firstly, the calculation of the P/F value requires blood gas analyses. In the clinical use of arterial indwelling catheters, daily care is difficult, and it is not easy to operate on some particular patients, such as newborns and elderly patients[7]. Secondly, arterial blood gas analyses cannot monitor the development of patients with ARDS in real-time, which makes doctors unable to adopt appropriate respiratory therapy strategies and delay the diagnosis and treatment of patients[8].

In recent years, in response to encountered problems in conducting blood gas analyses, researchers attempted to use the noninvasive parameter pulse oximetric saturation ($SpO_2$)/ $FiO_2$ (S/F) to estimate P/F, thereby achieving noninvasive identification of the severity of ARDS disease[9–11]. At this stage, the single $SpO_2$ parameter was mainly used, and there was a specific limit expected in relation to the range of $SpO_2$ ($SpO_2 \leq 97\%$). The traditional linear regression algorithm[11] was used to construct the prediction model, but the model identification effect was not ideal[10,12–16]. Additionally, it was challenging to provide accurate guidance for medical staff in the clinic[12,17].

Based on a review of the literature[9], we found that when a patient's condition changes, the patient's physiological parameters (such as heart rate, blood pressure, respiratory rate, etc.) will change at varying degrees, which provided ideas that assisted the investigation of the aims of our study.

In response to the problems listed above, we extracted a variety of noninvasive physiological parameters from ICU patients and explored the relevance of these parameters for the identification of the level of P/F ratio. An algorithmic model for identifying ARDS disease based on a variety of noninvasive parameters was established to provide medical staff with the reference basis for disease diagnosis. This model uses a feature selection algorithm and a cross-validation model to evaluate the recognition effects of four machine learning algorithms using different subsets of feature values.

Herein, we used a variety of evaluation indicators to assess the ability of different algorithms and feature subsets for ARDS disease identification. To further investigate the performance of machine learning algorithms, we used existing data to classify the ARDS disease using traditional linear regression models, and we discuss the various methods of development.

## 2. Materials

### 2.1 Data sources

Medical Information Mart for Intensive Care III (MIMIC-III, V1.4) is a large, freely available database comprising de-identified health-related data associated with over forty thousand patients who stayed in the critical care units of the Beth Israel Deaconess Medical Center between 2001 and 2012[18]. The database includes information, such as demographics, vital sign measurements obtained at the bedside, laboratory test results, procedures, medications, caregiver notes, imaging reports, and mortality records.

### 2.2 Patients and data collection

The patient diagnostic information was recorded in the MIMIC–III database. In the patient screening process, we combined the diagnostic information provided by the database and the Berlin definition to determine whether the enrolled patient was suffering from ARDS, thus ensuring the accuracy of the disease diagnosis. In combination with the Berlin definition and the disease diagnosis, we propose the following conditions: 1) determine whether the patient has a P/F < 300 on the first day of entering the ICU, 2) determine whether the patient underwent chest imaging during his/her presence in the ICU and whether the imaging report was

verified, 3) formulate a comprehensive judgment based on the patient's disease diagnosis information.

Combined with the above, we propose the corresponding patient selection criteria:

1. Choose the patients who first entered the ICU (if the patients entered the ICU multiple times, it may be likely that the patient conditions were more complicated and may have affected the identification result. In this study, we only used the data from the patients who entered the ICU for the first time)

2. The patient is older than 16 years old

3. The patient stayed in the ICU for more than 48 h

4. Mechanical ventilation was used during the presence of the patient in the ICU

5. P/F < 300 on the first day

This study extracted a variety of noninvasive physiological parameters of patients: demographics (age, gender, height, weight, body mass index (BMI), ethnicity), ICU information (ICU type, length of stay in ICU, admission type, in-hospital mortality), clinical measures ($SpO_2$, temperature, heart rate, blood pressure, Glasgow Coma Scale (GCS)), respiratory system (respiratory rate, tidal volume, minute ventilation volume, peak pressure, plateau pressure, mean air pressure, PEEP, $FiO_2$), and oxygenation index (P/F, S/F, Oxygenation Index (OI), Oxygenation Saturation Index (OSI)).

This study has paid more attention to the noninvasive physiological parameters of patients. Based on an extensive review of the literature combined with the actual recording parameters of patients in the database, the following noninvasive physiological parameters are finally used in the identification algorithm: $SpO_2$, temperature, heart rate, blood pressure, GCS, respiratory rate, tidal volume, minute ventilation volume, peak pressure, plateau pressure, mean air pressure, PEEP, $FiO_2$, S/F, OSI, and demographics (age, gender, BMI). Additionally, convert these parameters into 24 features for model training. The main purpose of this study was to identify ARDS by monitoring P/F values through a variety of noninvasive parameters. We used P/F as the outcome variable, $P/F \leq 300$ data points as positive samples, and $P/F > 300$ as negative samples.

In the process of extracting the physiological parameters for patients from the database, we also needed to extract the blood gas analyses outcomes obtained at a specific test time to ensure the accuracy of the identified results. However, this also caused considerable data losses. To avoid this problem, we allowed the use of data from the first two h following blood gas analyses in the data collection process as a substitute for the respective outcomes at the specific, desired test time.

## 3. Methods

This section provides an overview of the procedures described in the adopted methods, which are visually summarized in Fig 1. The dataset for this study was from the MIMIC–III database. After preprocessing the data, they were divided into a training (75%) and a test set (25%). In the model training process, we used the training dataset, used cross-validation to evaluate the identification performance of different feature subsets and algorithms, used the test set to verify the model, and compared it with the traditional algorithm.

### 3.1 Preprocessing

**3.1.1 Handling missing values.** In the process of collection of physiological parameters from patients, we have found that some physiological parameters were recorded at a lower

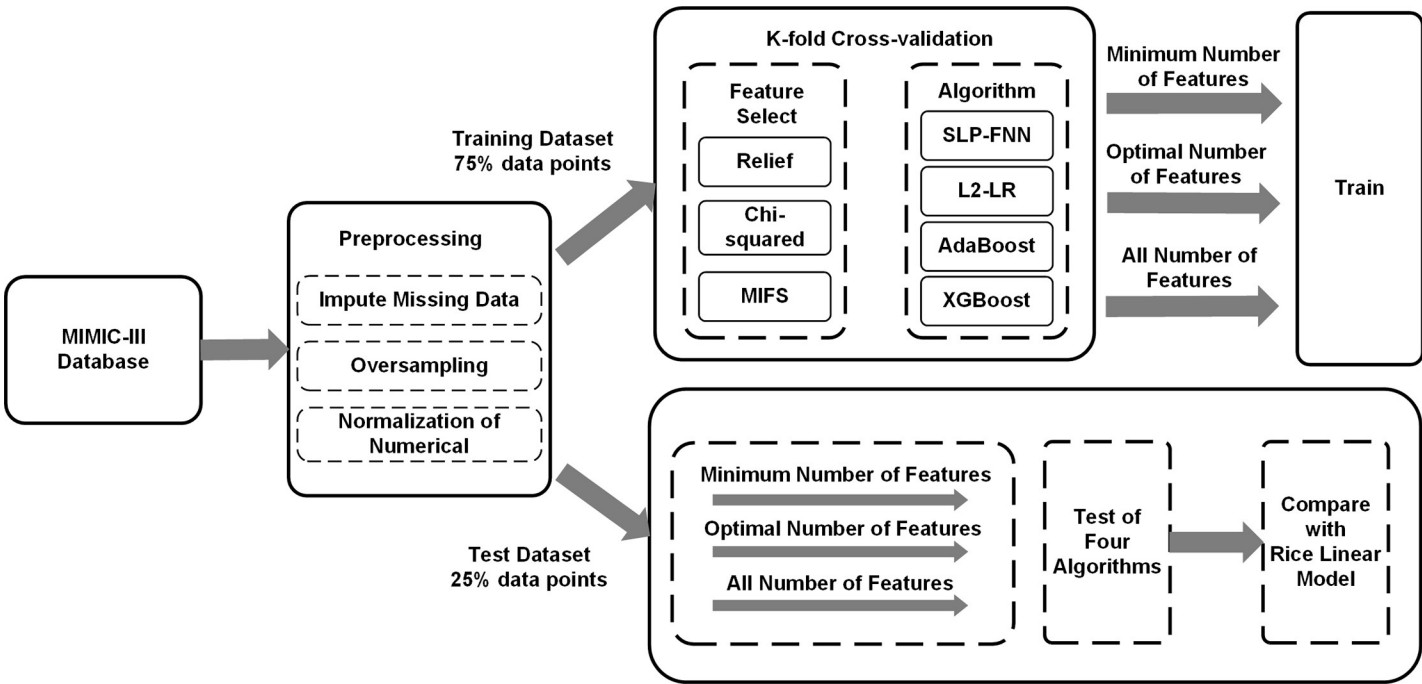

**Fig 1. Overview of the model design.** Summarize the overall process of the experiment. The raw data in the MIMIC-III database was preprocessed, and the data set was awakened and randomly grouped: 75% data was used for model training; the remaining 25% data was used for model testing, and comparative experiments were conducted to obtain the final experimental results.

frequency, such as the noninvasive blood pressure recordings, thus resulting in the absence of physiological data recordings at the time of the blood gas analyses. Fortunately, the patient's invasive blood pressure was continually monitored, which could provide data when noninvasive blood pressure data was lacking[19]. Therefore, the random forest was used to complement the patient's noninvasive blood pressure missing values, and other physiological parameters were imputed using k-nearest neighbor (k-NN)[20].

**3.1.2 Oversampling and normalization.** In the data preprocessing process, we found that use of P/F $\leq$ 300 to divide the dataset into positive and negative samples would result in an imbalance in the dataset. The use of unbalanced data for the machine learning algorithm training would result in a bias toward the larger sample size, which would make the generalization ability of the algorithm insufficient, and would affect the overall performance of the model. For the above reasons, we used the oversampling method to deal with the problem of data imbalance[21]. The current implementation methods of oversampling included random oversampling, the Synthetic Minority Oversampling Technique (SMOTE)[22], and the Adaptive Synthetic (ADASYN) sampling approach[23]. Random oversampling solves the problem of data imbalance by randomly sampling in the classes which are under-represented. SMOTE uses the similarities of under-represented samples in the feature space to generate new samples. The ADASYN solution was used to generate different numbers of new samples for different under-represented samples, based on the data distribution. It is an extension of SMOTE, but from the results, ADASYN tended to focus on some outliers. Based on the above analyses, we use SMOTE to deal with the problem of the data sample imbalance.

This study used a variety of physiological patient parameters each of which was associated with a different range of values. For most machine learning algorithms (such as neural networks), this situation would result in the slow learning of the algorithm, and would be easier to

achieve the local optimal solution, thereby affecting the training outcomes of the algorithm. Therefore, it was necessary to normalize the feature values of different orders of magnitude to the same order of magnitude [a, b]. We used feature scaling to standardize data in accordance with Eq (1):

$$X' = a + \frac{X - X_{\min}}{X_{\max} - X_{\min}}(b - a) \tag{1}$$

Where $X_{\min}$ represents the minimum and $X_{\max}$ represents the maximum value of an attribute. The motivation to use feature scaling was based on the robustness to very small standard deviations of features and the preservation of zero entries in the sparse data.

## 3.2 Feature selection

This study extracted a variety of information and physiological parameters of patients, and these parameters correspond to 24 features, but it is unclear which features yield a strong correlation with the identification of ARDS disease. Conversely, the performance of the supervised learning algorithm had a certain correlation with the number of input features, and the correlation between features and outcome variables. The purpose of feature selection was to identify a subset of features that optimized the algorithmic performance compared to the original feature set. There were three types of feature selection algorithms, namely, filter, wrapper, and embedded[24].

The filter method first selected the feature of the dataset and then trained the classifier. The feature selection process was independent of the subsequent classifier. In contrast to the filter method, the wrapper feature selection directly applied the performance of the classifier to be used as the evaluation criterion of the feature subset. In the case where more features existed, the computational overhead was usually much larger than that for the filter. The embedded method combined the evaluation feature importance with the model algorithm, and resulting in an increased correlation between the feature value selection results and the evaluation algorithm. In most cases, other algorithms were not applicable[25].

The filter method was associated with a small number of calculations, and the feature value selection result did not depend on the classification algorithm[24]. The filter method generally evaluates the importance of feature values in three ways: distance, dependency, and information. Based on this, we have selected three representative methods for these three aspects: relief, chi-squared, and mutual information.

**3.2.1 Relief-F.** The key idea of the Relief-F algorithm was to estimate the quality of attributes according to how well their values distinguished among instances that were close to each other[26]. For example, the quality estimation of the attribute j is shown in Eq (2), if sample $x_i^j$, belonged to class k, Relief–F first searched for the $x_{i,nh}^j$ (near-hit) of $x_i^j$ in the sample of class k, and then found $x_{i,nm}^j$ (near-miss) of x in each class other than the kth class.

$$\partial^j = \sum_i - diff\left(x_i^j, x_{i,nh}^j\right)^2 + \sum_{l \neq k}\left(p_l \times diff\left(x_i^j, x_{i,nm}^j\right)^2\right) \tag{2}$$

**3.2.2 Chi-squared.** The chi-squared was based on the $\chi^2$ statistic and consisted of two phases. The first phase began with a high significance level for all numeric attributes for discretization. Each attribute was sorted according to its values[27]. The following steps were then performed: 1) calculate the $\chi^2$ value in accordance with Eq (3) for every pair of adjacent intervals, and 2) merge the pair of adjacent intervals with the lowest $\chi^2$ value. Eq (3) was used for

computing the $\chi^2$ value in accordance to

$$\chi^2 = \sum_{i=1}^{2} \sum_{j=1}^{k} \frac{(A_{ij} - E_{ij})^2}{E_{ij}} \tag{3}$$

Where $A_{ij}$ denotes the number of patterns in the $i$th interval and $j$th class, and $E_{ij}$ is the expected frequency of $A_{ij}$.

**3.2.3 Mutual information.** Mutual information based feature selected(MIFS) was used to measure the amount of information shared between the two features[28]. The mutual information $I(X,Y)$ between two variables X and Y was expressed as

$$I(X, Y) = \int P_{xy}(x, y) \log_2 \left( \frac{P_{xy}(x,y)}{P_x(x)P_y(y)} \right) dx dy \tag{4}$$

**3.2.4 Rank aggregation.** To ensure the stability of feature selection, we used a combination of filter feature selection methods[25]. The results of the three algorithms were not in a uniform numerical range, and it is not convenient to evaluate the importance of the features. In order to make the three methods equally important, we normalized the three results[29]. The rank aggregation method was formulated based on Eq (5).

$$R_j = \frac{1}{n} \sum_{i=1}^{n} \frac{F_{i,j}}{F_{i,\max} - F_{i,\min}} \tag{5}$$

Where $F_i$ is the result of different filter feature selection methods, and $R_j$ is the final rank score for the jth feature.

## 3.3 Classification algorithms

This study designed an algorithm that combined feature selection with multiple classification algorithms, used a 10-fold cross-validation model, trained classifiers for different feature subsets, and selected the optimal combination of feature subsets and classifiers, and achieved the identification of the ARDS. This section presents an abridged description of the four classifiers selected for this study.

**3.3.1 L2 regularized logistic regression (L2–LR).** In order to prevent overfitting of the classification algorithm, a regularization term was added to the traditional logistic regression cost function $J(w,b)$. Since the feature selection used the external filter method, this study used L2 regularization to avoid the situation where the $L_1$ regularization caused the weight to be sparse[30]. Furthermore, $\lambda$ is the regularization parameter used to control the weight $w$.

$$\hat{y} = \sigma(w^T x + b) \tag{6}$$

$$J(w, b) = \min_{w,b} \left\{ -\frac{1}{m} \sum_{i=1}^{m} [y_i \ln \hat{y}_i + (1 - y_i) \ln(1 - \hat{y}_i)] + \frac{\lambda}{2n} \sum_{w} \|w\|_2 \right\} \tag{7}$$

Where $\hat{y}_i$ is defined in accordance to Eq (6), m is the number of samples, and n is the number of features.

**3.3.2 Artificial neural network.** This study used a single hidden layer feedforward neural network (SLP–FNN). According to the number of features and the outcome variables, the following network structure was designed. Specifically, the number of neurons in the input layer was 24, the number of neurons in the hidden layer was 23, and the number of neurons in the output layer was two. In order to quickly iterate and train the network, we used a stochastic gradient descent algorithm to optimize the parameters of the network, while we concurrently

used adaptive learning rates. Selecting the rectified linear unit function as the activation function we could effectively prevent the occurrence of gradient disappearance. To prevent overfitting in the network training, we used the L2 regularization term. The principle is the same as that described in subsection 3.3.1.

**3.3.3 AdaBoost.** The AdaBoost algorithm is a two-class learning method in which the model is an additive model, the loss function is an exponential function, and the learning algorithm is a forward step-by-step algorithm. The specific idea of AdaBoost was to increase the weights of samples that had been misclassified by the previous round of weak classifiers, and to reduce the weights of those samples that were correctly classified[31]. As a result, the data that were not correctly classified were more concerned by the latter round of weak classifiers owing to their increased weight. Herein, $G_m(x)$ is a weak classifier, and $\alpha_m$ indicates the importance of $G_m(x)$ in the final classifier, Eq (8) is a mathematical description of the forward distributed algorithm, and Eq (9) is the final classifier constructed based on Eq (8).

$$f(x) = \sum_{m=1}^{M} \alpha_m G_m(x) \tag{8}$$

$$G(x) = sign(f(x)) = sign\left(\sum_{m=1}^{M} \alpha_m G_m(x)\right) \tag{9}$$

**3.3.4 XGBoost.** XGBoost is a scalable machine learning system for tree boosting. The impact of the system has been extensively recognized in a number of machine learning and data mining challenges[32].

$$\tilde{L}^{(t)} = \sum_{i=1}^{n} l(y_i, \hat{y}_i^{(t-1)} + f_t(x_i)) + \Omega(f_t) \tag{10}$$

$$\Omega(f_t) = \gamma T + \frac{1}{2}\lambda \sum_{j=1}^{T} w_j^2 \tag{11}$$

Herein, $\tilde{L}^{(t)}$ is a differentiable convex loss function that measures the difference between the prediction $\hat{y}_i$ and the target $y_i$. The second term $\Omega(f_t)$ penalizes the complexity of the model. The additional regularization term helps to smooth the final learned weights to avoid overfitting. Moreover, $\gamma$ and $\lambda$ are the regularization parameters used to control regularization terms.

**3.3.5 Traditional noninvasive classification method.** Previous studies on the use of the noninvasive parameter identification ARDS focused on the use of a single parameter S/F to fit the P/F value. This study used the linear regression model proposed by Rice et al[11]. The model used adult $SpO_2$ values ($SpO_2 < 97\%$) to fit the P/F values, thus enabling continuous monitoring of the patient's P/F values using noninvasive parameters. The Rice Linear Model is shown in Eq (12).

$$S/F = 64 + 0.84 \times (P/F) \tag{12}$$

The noninvasive parameter S/F is used to obtain the predicted P/F value according to Eq (12) so as to classify the severity of ARDS disease, and to obtain the classification result of the traditional algorithm.

## 3.4 Performance metrics

According to the diagnostic definition of ARDS disease, P/F $\leq$ 300 is ARDS. According to this standard, the sample is divided into positive and negative results. Table 1 describes the relationship between the real category and the identification category.

**Table 1. The relationship between real categories and recognition results.**

| Predicted class | Actual class | |
|---|---|---|
| | **Positive(P/F≤300)** | **Negative(P/F>300)** |
| **Positive** | True positive (TP) | False positive (FP) |
| **Negative** | False negative (FN) | True negative (TN) |

We measured the classification performance based on the average of AUC, and the accuracy (ACC), sensitivity (SEN), specificity (SPE), and balanced error rate (BER), as defined by Eqs (13)–(16), respectively.

$$ACC = \frac{TN + TP}{TP + FP + FN + TN} \qquad (13)$$

$$SEN = \frac{TP}{TP + FN} \qquad (14)$$

$$SPE = \frac{TN}{TN + FP} \qquad (15)$$

$$BER = 1 - \frac{1}{2}(SEN + SPE) \qquad (16)$$

BER is a balanced metric that equally weights errors in SEN and SPN. We used the BER index to select the optimal feature subset based on a 10-fold cross-validation model[33]. For each algorithm, under the different feature subsets, the smallest mean BER was chosen as the optimal feature subset (the minimum BER subset) of this algorithm[29]. The search algorithm for optimal feature subsets is summarized in Algorithm 1 (Fig 2). According to the results of this algorithm, the minimum feature subset of the algorithm was found within the BER standard deviation of the optimal feature subset. At the same time, the two cases presented above were compared with all of their features to select the optimal identification result.

## 4. Results

We identified 8702 patients who met our inclusion criteria from a total of 46476 patients enrolled in the MIMIC–III database. Fig 3 is a flowchart outlining the patient selection and detailing the number of patients and the data selection process. There were 6601 patients (148414 data points) in the training set and 2101 patients (47352 data points) in the test set.

The demographics and utilization characteristics are summarized in Tables 2 and 3. Table 2 summarizes the demographic information of patients. The training set has a consistent patient distribution with the test set. In the training set, the patients were hospitalized in different intensive care units: CSRU (2231, 33.8%), MICU (1851, 28.4%), SICU (927, 14.04%), TSICU (904,13.09%), and CCU (688, 10.42%), the average age of patients was 65.14. The majority of the patients were male (58.64%). The patient in-hospital mortality rate was 16.34%. Table 3 summarizes the distribution of physiological parameters of patients classified in the training and test sets. As observed, there is a large difference between the positive samples (P/F ≤ 300) and the negative samples (P/F > 300) within the same dataset. For the training set and the test set, the two datasets were randomly grouped and had a common distribution. There is no significant difference in the dataset.

**Algorithm 1**. MIN–BER–FS algorithm

**Input:**     Training set  $D = \left\{ (x_1, y_1), (x_2, y_2), \cdots, (x_m, y_m) \right\}$.

Sort the feature according to the combined filter score s

Initialization:  $I = \left\{ D(:, s), y \right\}$

**Process:**

(1)     $BER_{\min} = \infty, j = 1, index_{\min BER} = 0$

(2)     **Repeat:**

(3)     $I^* = \left\{ D(:, s[1:j]), y \right\}$

(4)     Ten-fold cross-validation for the training set  $I^*$  to determine the BER of

$I^* : BER_{k, I^*} (k = 1, 2, \cdots, 10)$

(5)     $meanBER_{I^*} = 1/10 \sum_{k=1}^{10} BER_{k, I^*}$

(6)     **if**  $meanBER_{I^*} < BER_{\min}$  **then**

(7)         Update current  $BER_{\min}$  to  $meanBER_{I^*}$

(8)     $index_{\min BER} = j_{index}$

(9)     **end if**

(10)     $j = j + 1$

(11)     **Until**  $j > \mathrm{len}(s)$

**Output:**     $BER_{\min}, index_{\min BER}$

**Fig 2. Algorithm 1.** MIN–BER–FS algorithm.

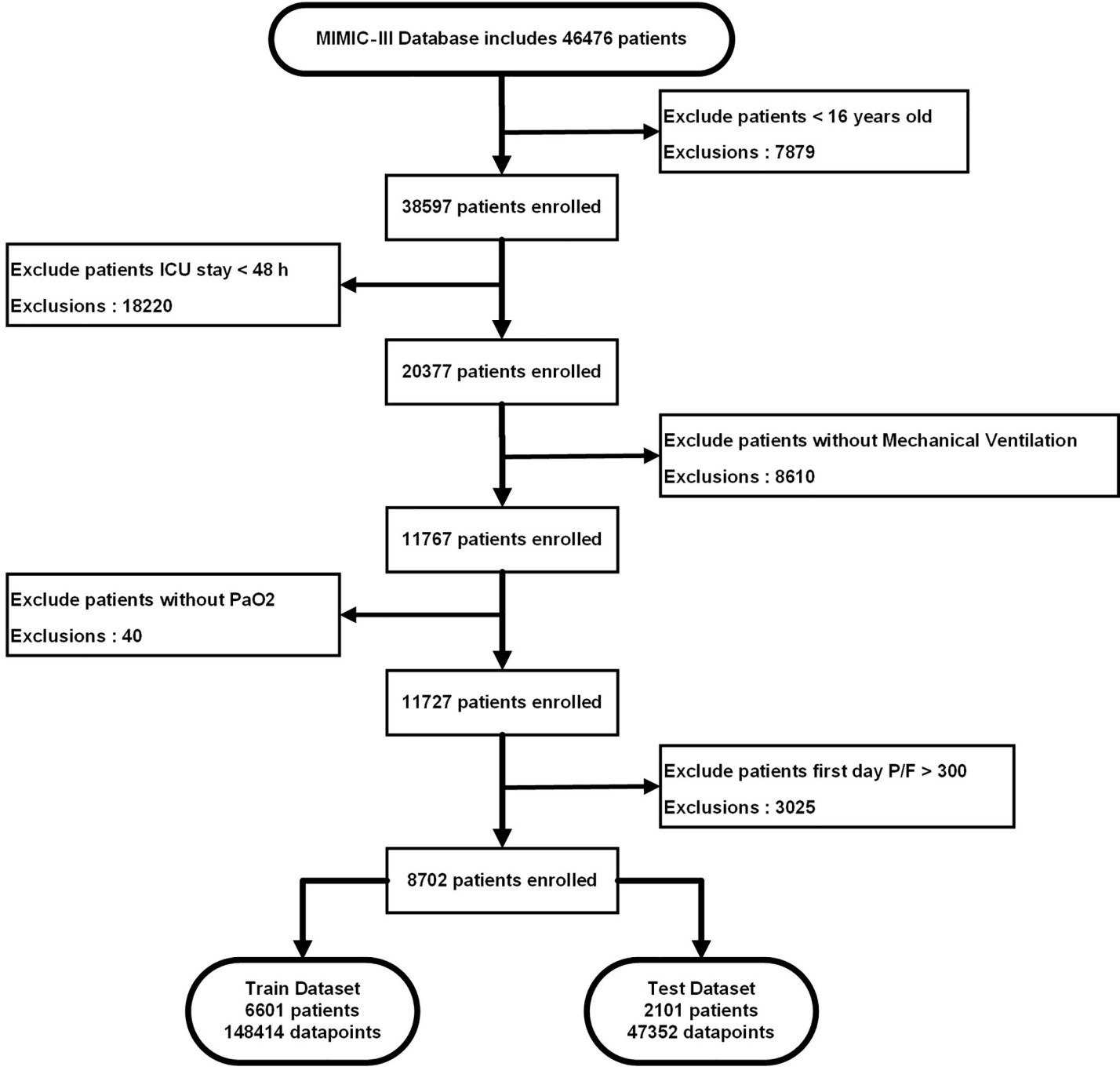

**Fig 3. Flow diagram for patient selection.** According to the ARDS diagnostic criteria, the appropriate enrolled population was selected from more than 40,000 patients in the MIMIC-III database, and 8702 eligible patients were finally included, and the data sets were randomly divided into training sets and test sets.

## 4.1 Feature selection result

Table 4 represents the normalized values of the scores provided by the three filter methods under consideration. The importance of the features in this study is relative to the level of oxygenation index. The closer the value is to one of the scores, the more relevant the feature is. The MIFS criterion showed that a number of parameters were relevant, while the Relief–F and

**Table 2. Patient demographics in training and test sets (ICU: Intensive Care Unit, CSRU: Cardiac Surgery Recovery Unit, MICU: Medical Intensive Care Unit, CCU: Coronary care unit, SICU: Surgical intensive care unit, TSICU: Trauma Surgical Intensive Care Unit).**

| Variables | Training dataset | Test dataset |
|---|---|---|
| | n = 6601 | n = 2101 |
| Age(year)[*] | 65.14±16.16 | 64.89±16.00 |
| Gender, n (%) | | |
| Female | 2730(41.36) | 845(40.22) |
| Male | 3871(58.64) | 1256(59.78) |
| BMI(kg/m$^2$)[*] | 29.52±7.81 | 29.91±8.12 |
| Length of stay in ICU (days)[+] | 5.04(3.08–9.89) | 4.98(3.04–10.15) |
| Mortality, n (%) | 1080(16.34) | 331(15.75) |
| ICU type, n (%) | | |
| CSRU | 2231(33.80) | 741(35.27) |
| MICU | 1851(28.04) | 583(27.75) |
| SICU | 927(14.04) | 287(13.66) |
| TSICU | 904(13.09) | 266(12.66) |
| CCU | 688(10.42) | 224(10.66) |
| Admission type, n (%) | | |
| Emergency | 5252(79.56) | 1681(80.01) |
| Elective | 1349(20.44) | 418(19.90) |
| Ethnicity, n (%) | | |
| White | 4568(69.11) | 1485(70.68) |
| Asian | 143(2.16) | 37(1.76) |
| Black | 372(5.63) | 133(6.33) |
| Hispanic | 198(3.00) | 59(2.81) |
| Other | 1320(19.97) | 387(18.42) |

[*] Data are mean ± SD

+ median (interquartile range)

chi-squared tests were more conservative, and indicated that the $SpO_2$ and S/F were the more important and relevant features. The ranking of the final features is listed in Table 3. This combined score was calculated based on Eq (5). According to the combined score, $SpO_2$ is clearly more relevant than the rest of the parameters. Furthermore, $SpO_2$, S/F, $FiO_2$, and PEEP, are also likely to be highly relevant features.

## 4.2 Algorithmic evaluation

Using the training dataset, the 10-fold cross-validation methods were used to evaluate the performance of the four algorithms. According to the feature ranking results in Table 4, the features were substituted into the four algorithms in turn, the BER of each algorithm was used to select the feature subset, and the algorithm effect was compared based on AUC. As shown in Fig 4, the BER of the four algorithms change as a function of the number of features, and the average BER results of the four classification algorithms are listed for different feature subsets. The gray area represents the standard deviation of the BER. The red triangle and green dot marks and their corresponding numbers represent the minimum feature and the optimal feature subsets, respectively. We found that the BER of the four algorithms decreased

**Table 3. Patient characteristics in training and test sets (Nisbp: Noninvasive systolic blood pressure, Nidbp: Noninvasive diastolic blood pressure, Nimbp: Noninvasive mean blood pressure, OI: ($FiO_2 \times Mean\ air\ pressure$)/$PaO_2$, OSI: ($FiO_2 \times Mean\ air\ pressure$)/$SpO_2$.**

| Variables | Training dataset | | Test dataset | |
|---|---|---|---|---|
| | (n = 148414) | P/F>300(n = 34647) | (n = 47352) | P/F>300(n = 10972) |
| | P/F≤300(n = 113767) | | P/F≤300(n = 36380) | |
| $SpO_2$* | 96.57±3.70 | 98.72±2.93 | 96.56±3.58 | 98.68±2.93 |
| S/F* | 183.31±50.02 | 218.80±54.04 | 184.08±49.88 | 219.72±53.87 |
| OSI* | 7.90±5.04 | 5.37±2.92 | 7.79±4.91 | 5.28±2.88 |
| $PaO_2$* | 102.65±37.30; | 200.74±91.84 | 102.00±36.28 | 198.74±89.74 |
| P/F* | 188.98±61.91 | 410.72±142.58 | 188.78±61.23 | 408.82±141.58 |
| OI* | 8.21±6.51 | 2.80±1.22 | 8.09±6.13 | 2.77±1.17 |
| $FiO_2$(%)* | 57.64±19.21 | 49.23±17.98 | 57.31±19.00 | 49.03±18.08 |
| Temperature (˚C)* | 37.14±0.88 | 36.93±0.94 | 37.16±0.88 | 36.95±0.94 |
| Respiratory rate(b/min)+ | 20(16–25) | 18(14–23) | 20(16–25) | 18(14–23) |
| Tidal volume(mL)* | 551.39±116.71 | 555.89±115.70 | 550.58±114.49 | 556.62±115.32 |
| Tidal volume(mL/kg)* | 6.72±2.02 | 7.14±2.05 | 6.64±2.10 | 7.21±2.22 |
| Minute ventilation volume(mL/min)* | 10.69±3.31 | 9.56±3.14 | 10.69±3.32 | 9.52±3.05 |
| Peak pressure(cmH$_2$O)* | 26.62±7.86 | 24.46±6.99 | 26.50±7.76 | 24.36±6.78 |
| plateau pressure(cmH$_2$O)* | 22.82±5.60 | 20.67±4.59 | 22.69±5.56 | 20.58±4.38 |
| Mean air pressure(cmH$_2$O)* | 12.75±5.15 | 10.68±3.88 | 12.66±5.00 | 10.53±3.62 |
| PEEP(cmH$_2$O)+ | 7(5–10) | 5(5–7) | 6(5–10) | 5(5–7.6) |
| Heart rate(bpm)+ | 88(78–101) | 86(75–97) | 88(77–100) | 87(77–97) |
| Nisbp(mmHg)* | 115.38±15.44 | 116.49±15.70 | 115.36±15.38 | 115.61±15.36 |
| Nidbp(mmHg)* | 56.00±9.56 | 56.92±9.85 | 56.20±9.42 | 56.37±9.78 |
| Nimbp(mmHg)* | 73.12±10.02 | 74.21±10.36 | 73.26±9.97 | 73.53±10.01 |
| GCS+ | 12(6–14) | 10(5–13) | 12(7–14) | 10(4–13) |

* Data are mean ± SD

+ median (interquartile range)

considerably when the first five features were added to the model, but as the features were added gradually, the BER decreased slowly.

For SLP–FNN, L2–LR, and AdaBoost, almost all feature training models were used to achieve minimum BER. Compared to the first three algorithms, XGBoost achieved the minimum BER in the 13th feature. As the number of features increased, the BER appeared to increase. We selected the smallest number of features for which the mean BER was within one standard error of the minimum BER (subset selection threshold). According to this standard, we found the optimal and smallest feature subset of the four algorithms: L2–LR (24, 18), SLP–FNN (24, 20), AdaBoost (23, 21), XGBoost (12, 6).

## 4.3 Performance of classification algorithms

**4.3.1 Training dataset.** Based on the selected features, we obtained the minimum, optimal feature subset for the training set. We used training data for the minimum, optimal, and all feature subsets. Four classification algorithms were trained using 10-fold cross-validation. The results are shown in Table 5.

By comparing the results of the optimal and minimum feature subsets, we found that the minimum feature subset was determined using the minimum BER and standard deviation, but with the use of fewer feature quantities (reducing certain data information). However,

**Table 4. Physiological parameter scores and rankings for different feature selection methods.**

| Feature | MIFS | Relief-F | Chi-squared | Aggregation |
|---|---|---|---|---|
| SpO$_2$ | 0.9517 | 1.0000 | 1.0000 | 0.9839 |
| S/F | 0.7484 | 0.7314 | 0.8154 | 0.7651 |
| FiO$_2$ | 0.9610 | 0.4044 | 0.3971 | 0.5875 |
| PEEP | 0.8666 | 0.3412 | 0.2989 | 0.5023 |
| Mean air pressure | 0.9316 | 0.2589 | 0.1667 | 0.4524 |
| Respiratory rate | 0.9569 | 0.3014 | 0.0761 | 0.4448 |
| Plateau pressure | 0.9078 | 0.2549 | 0.1666 | 0.4431 |
| GCS (eye) | 0.9676 | 0.2820 | 0.0499 | 0.4332 |
| GCS (motor) | 0.9577 | 0.2056 | 0.0375 | 0.4003 |
| TV/kg | 1.0000 | 0.1502 | 0.0475 | 0.3992 |
| GCS (verbal) | 0.9367 | 0.2157 | 0.0365 | 0.3963 |
| Gender(female) | 0.9660 | 0.0000 | 0.1466 | 0.3709 |
| Peak pressure | 0.8401 | 0.1775 | 0.0782 | 0.3652 |
| OSI | 0.3259 | 0.2795 | 0.4875 | 0.3643 |
| GCS | 0.8870 | 0.1582 | 0.0399 | 0.3617 |
| Heart rate | 0.8950 | 0.1587 | 0.0228 | 0.3588 |
| Gender(male) | 0.9772 | 0.0000 | 0.0963 | 0.3578 |
| Nidbp | 0.9354 | 0.0829 | 0.0030 | 0.3405 |
| Minute ventilation volume | 0.6954 | 0.1859 | 0.1206 | 0.3340 |
| Nisbp | 0.9124 | 0.0778 | 0.0003 | 0.3302 |
| Temperature | 0.8066 | 0.1290 | 0.0189 | 0.3182 |
| Nimbp | 0.8592 | 0.0666 | 0.0027 | 0.3095 |
| Age | 0.0000 | 0.4365 | 0.0000 | 0.1455 |
| BMI | 0.2376 | 0.1515 | 0.0422 | 0.1438 |

there was no significant decline in AUC. Fig 5 shows the AUC results for each algorithm with the use of different feature subsets.

### 4.3.2 Test dataset

The test set was completely independent of the data of feature selection and model training. When the training set was used to test the performance of the algorithm, we added a traditional noninvasive identification algorithm to compare the traditional and the machine learning algorithm. The final result is shown in Table 6.

The ROC curves of the five algorithms are shown in Fig 6. Based on the results, we show that the overall performance of the traditional algorithm exhibits a specific gap with respect to the machine learning algorithm. The AUC (0.7354) of the Rice Linear Model is much lower than the AUC of L2–LR under the minimum feature subset (0.8156).

In this study, we analyze the classification ability of features, use filter methods to sort the importance of features, and use MIN–BER–FS algorithm to find the optimal feature subset and the minimum feature subset. In the algorithmic evaluation, we compare the experimental results of the minimum, optimal, and all feature subsets, which can reflect the ability of the algorithm to mine information from different aspects. In the process of feature traversal, it can also be said that the feature importance is consistent with the feature sorting experiment results. From the results, XGBoost has the best results under the optimal subset. At the same time, XGBoost achieved better results than the other three algorithms in the minimum feature

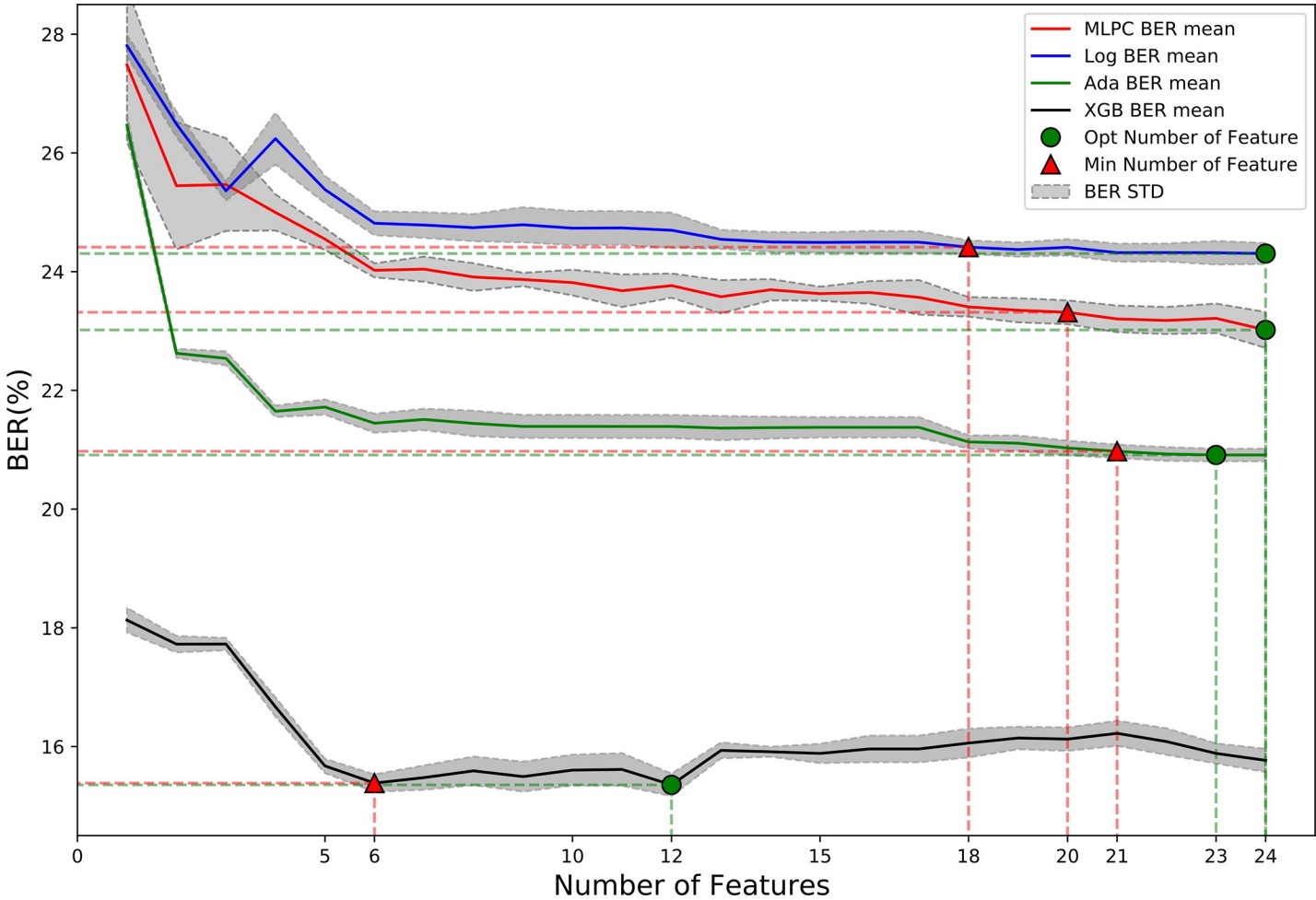

**Fig 4. Feature selection based on the four methods discussed in this study.** The X-axis is the feature number, the y-axis is the BER average of the ten-fold cross-validation, and the gray shaded area is the BER standard deviation of the ten-fold cross-validation under a specific feature subset. The figure shows the trend of BER changes of the four algorithms in the process of adding features step by step. The position of the green circle is the optimal feature subset of the algorithm, and the red triangle is the smallest feature subset.

subset (only the first six features). Observing the variation of the standard deviation during the training process, we can see that AdaBoost has the best stability.

## 5. Discussion

In this study, a novel identification algorithm was presented that combined multiple noninvasive physiological parameters with machine learning algorithms to estimate P/F ratio levels. First, we used the MIMIC–III database to extract the $SpO_2$, $PaO_2$, and $FiO_2$ that were commonly used to identify ARDS. At the same time, we extracted a variety of other noninvasive physiological parameters relevant to the patient. In terms of feature selection, the filter method was selected, and its feature selection was independent of the subsequent models. We used a variety of feature selection algorithms (Relief, chi-squared, MIFS) to filter the features, combined with the rank aggregation method, to obtain the final feature ranking results[25]. In the process of designing the ARDS identification algorithm, we used the cross-validation model to evaluate the average BER of the four algorithms (L2–LR, SLP–FNN, AdaBoost, XGBoost) in the optimal feature subset and the minimum feature subset according to the results of feature

**Table 5. Identification results of the four algorithms on the training set for different feature subsets.**

| Feature subset | Algorithm | Result | | | | | | Number of Features |
|---|---|---|---|---|---|---|---|---|
| | | SEN (%) | SPN (%) | PPV (%) | NPV (%) | ACC (%) | AUC | |
| **Minimum Number of Features** | SL–FNN | 80.61±2.79 | 72.76±2.96 | 74.80±1.39 | 79.04±1.76 | 76.68±0.20 | 0.8429±0.0013 | 20 |
| | L2–LR | 79.72±0.18 | 71.45±0.10 | 73.63±0.09 | 77.90±0.16 | 75.59±0.12 | 0.8249±0.0017 | 18 |
| | AdaBoost | 81.47±0.18 | 76.59±0.23 | 77.68±0.16 | 80.52±0.14 | 79.03±0.11 | 0.8687±0.0019 | 21 |
| | XGBoost | 83.02±1.39 | 86.21±1.17 | 85.77±0.87 | 83.56±0.91 | 84.61±0.15 | 0.9192±0.0054 | 6 |
| **Optimum Number of Features** | SL–FNN | 81.56±2.77 | 72.41±3.36 | 74.79±1.63 | 79.79±1.73 | 76.98±0.30 | 0.8464±0.0023 | 24 |
| | L2–LR | 79.50±0.30 | 71.89±0.14 | 73.88±0.14 | 77.81±0.26 | 75.70±0.18 | 0.8268±0.0017 | 24 |
| | AdaBoost | 81.58±0.16 | 76.59±0.21 | 77.71±0.15 | 80.62±0.13 | 79.09±0.11 | 0.8694±0.0019 | 23 |
| | XGBoost | 84.96±1.64 | 83.97±1.53 | 84.15±1.07 | 84.84±1.13 | 84.65±0.19 | 0.9282±0.0068 | 12 |
| **All Features** | SLP–FNN | 81.56±2.77 | 72.41±3.36 | 74.79±1.63 | 79.79±1.73 | 76.98±0.30 | 0.8464±0.0023 | 24 |
| | L2-LR | 79.50±0.30 | 71.89±0.14 | 73.88±0.14 | 77.81±0.26 | 75.70±0.18 | 0.8268±0.0017 | 24 |
| | AdaBoost | 81.71±0.21 | 76.23±0.12 | 77.69±0.19 | 80.57±0.13 | 79.05±0.11 | 0.8691±0.0020 | 24 |
| | XGBoost | 83.18±0.44 | 85.28±0.36 | 84.97±0.28 | 83.53±0.33 | 84.23±0.20 | 0.9241±0.0017 | 24 |

sorting to comprehensively consider the number of features and the identification results, thereby allowing the choice of the most suitable combination[29,33,34]. Conversely, selection of the minimum number of features implied the elimination of features that were insensitive to the accuracy of identification, simplification of the difficulty of the identification algorithm in actual use, and saving computation time. Conversely, the accuracy of the identification algorithm cannot be sacrificed.

Regarding the noninvasive identification related research on the severity of ARDS, most of the current research is concerned on the relationship between S/F and P/F[9,10,12,35]. S/F and P/F did exhibit strong correlations, but the use of S/F for regression analyses alone led to a large error in the classification of the severity of ARDS disease[6], and there was a range of restrictions on $SpO_2$ ($SpO_2 < 97\%$)[10,11]. Some researchers have found that P/F was affected by some other parameters, such as the possible connection to a ventilator, or the modification

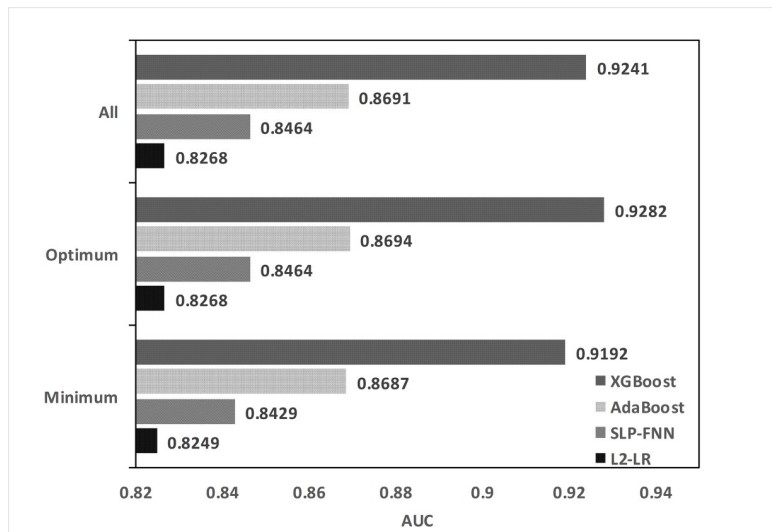

**Fig 5. AUC of the four tested algorithms on the training set for different feature subsets.** Based on the feature selection experiment, the training data set is used to study the recognition performance of four machine learning algorithms under different feature subsets.

**Table 6. Identification results of four algorithms on test sets for different feature subsets.**

| Feature Select | Algorithm | Result | | | | | | |
|---|---|---|---|---|---|---|---|---|
| | | SEN (%) | SPN (%) | PPV (%) | NPV (%) | ACC (%) | BER (%) | AUC |
| **Minimum Number of Features** | SL–FNN | 77.91 | 73.30 | 46.81 | 91.67 | 74.36 | 24.40 | 0.8307 |
| | L2–LR | 78.62 | 71.69 | 45.39 | 91.54 | 73.16 | 25.14 | 0.8156 |
| | AdaBoost | 74.06 | 75.97 | 48.17 | 90.66 | 75.52 | 24.99 | 0.8376 |
| | XGBoost | 81.68 | 87.92 | 87.11 | 82.75 | 84.80 | 15.20 | 0.9086 |
| **Optimum Number of Features** | SL–FNN | 75.10 | 76.17 | 48.73 | 91.03 | 75.92 | 24.36 | 0.8310 |
| | L2–LR | 77.74 | 72.03 | 45.60 | 91.48 | 73.35 | 25.11 | 0.8161 |
| | AdaBoost | 74.00 | 75.93 | 48.11 | 90.64 | 75.58 | 25.04 | 0.8434 |
| | XGBoost | 84.03 | 87.75 | 87.28 | 84.61 | 85.89 | 14.11 | 0.9128 |
| **All Features** | SLP–FNN | 75.10 | 76.17 | 48.73 | 91.03 | 75.92 | 24.36 | 0.8310 |
| | L2-LR | 77.74 | 72.03 | 45.60 | 91.48 | 73.35 | 25.11 | 0.8161 |
| | AdaBoost | 74.00 | 75.93 | 48.11 | 90.64 | 75.58 | 25.04 | 0.8334 |
| | XGBoost | 81.23 | 87.45 | 88.38 | 81.19 | 84.01 | 15.87 | 0.8957 |
| Rice Linear Model | | 39.45 | 72.87 | 69.73 | 77.49 | 70.67 | 48.16 | 0.7738 |

of ventilator-related settings (PEEP, FiO$_2$, Minute ventilation volume, etc.)[9]. At the same time, when the P/F changed, some physiological parameters of the patient (such as heart rate, respiratory rate, etc.) also changed[7]. Based on the above analyses, this study considered a variety of noninvasive physiological parameters obtained from patients. In the feature selection method design, we did not use an algorithm alone, but chose a variety of algorithms and integrated the results to prevent the selection of a single sorting algorithm to make the feature sorting less accurate. We selected three representative methods in accordance with distance, dependency, and information, and normalized the three feature ranking results to calculate the final feature ranking outcome. Compared to a single method, the sorting result is more stable and reliable.

Table 5 shows the results of the training set. L2–LR achieved a minimum BER when all the features were used, thus yielding an AUC = 0.8268. The neural network also reached the minimum BER when all the features were used, and its recognition performance was slightly better than L2–LR, thus yielding an AUC = 0.8464. Both AdaBoost and XGBoost were lifting tree algorithms. The identification results of the lifting tree algorithm were better than the logistic regression and neural networks (single hidden layer). When AdaBoost used 23 features, the BER outcome was the smallest, and yielded AUC = 0.8694. When XGBoost used 12 features, the BER outcome was minimal and yielded an AUC = 0.9282. Using the average minimum BER to find the minimum feature subset, it was found that reducing the number of features to a certain extent did not affect the recognition performance of the algorithm. In this respect, the advantage of XGBoost is obvious. With the use of six features, the accuracy rate only dropped by 0.42%. Combined with Fig 4, we found that the first six features (SpO$_2$, S/F, FiO$_2$, PEEP, mean air pressure, respiratory rate) contributed considerably to the identification algorithm, and the BER decrease was more distinct. After the addition of the features, the BER decreased gradually.

In the test set, we introduced a traditional linear regression algorithm[11] to evaluate the recognition performance of the classification algorithm. The performance of the four algorithms on the test set was basically consistent with the results of the training set, and yielded good generalization ability for the single-center independent dataset. The Rice Linear Model yielded an AUC = 0.7738 and ACC = 70.67%, which are far from the corresponding results elicited based on the machine learning algorithm. According to the literature published by

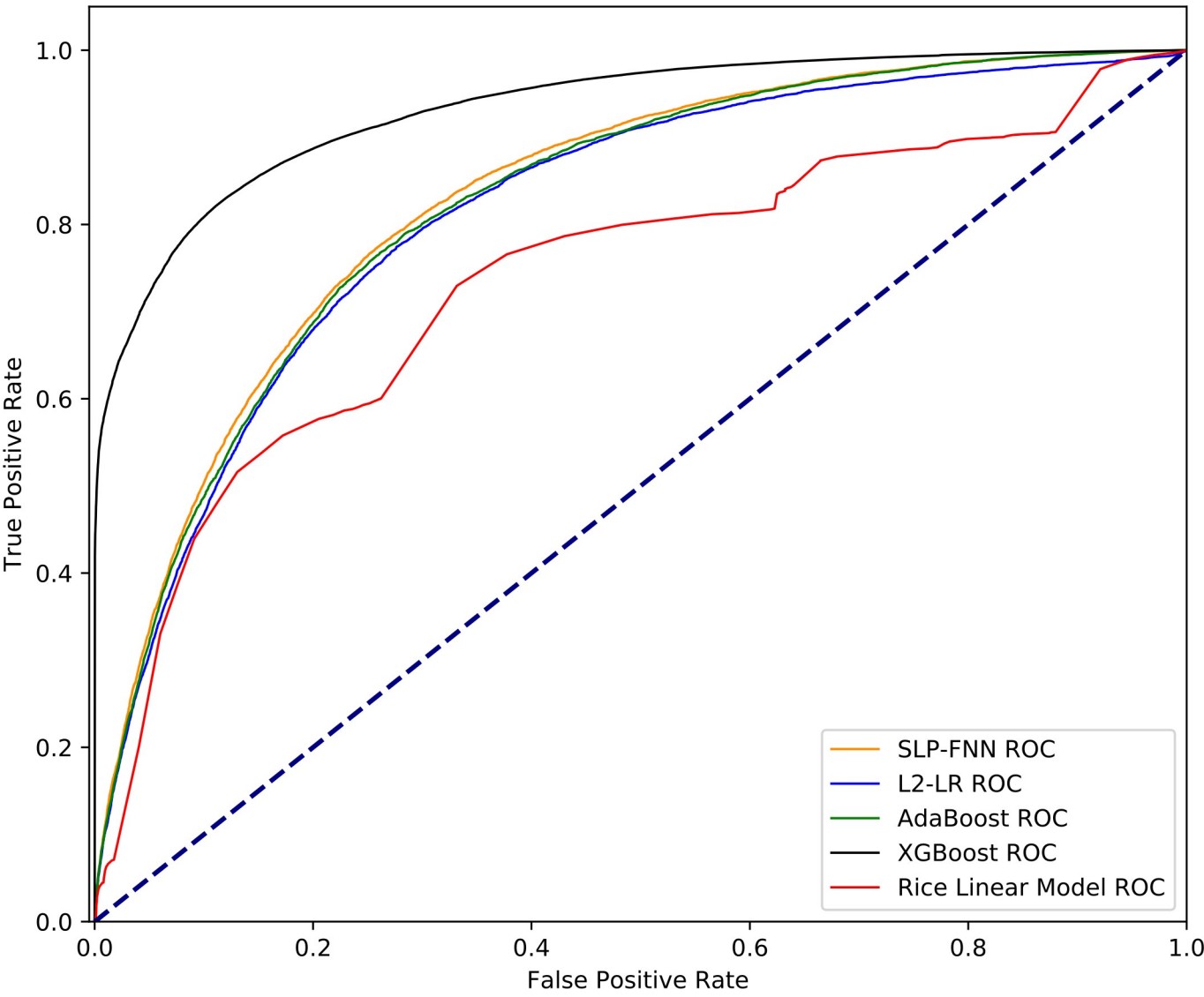

**Fig 6. ROC curves of the applications of the four algorithms studied herein on the test dataset.** According to the experimental results of the training set, the four machine learning algorithms and the Rice linear model are used to identify the performance of the minimum feature subset on the test set data, and the ROC curve of each algorithm is drawn.

Rice in 1994, we know that the Rice Linear Model method was based on the premise of $SpO_2$ < 97%[11]. This study directly used the model for the existing data (and the range of $SpO_2$ without limitations), and some deviation was expected. In actual clinical problems, there are often some ARDS patients with $SpO_2$ > 97%, but their P/F ≤ 300. In this case, Rice Linear Model cannot provide doctors with auxiliary diagnosis decisions, and our algorithm model can overcome the shortcomings of traditional methods.

The application scenario for this algorithm is as shown in S1 Fig. The patient's oxygenation level (≤300 or >300) was identified by collecting mechanical ventilation parameters and physical signs to assist the physician in the diagnosis of ARDS without blood gas analysis. For patients who have been diagnosed as ARDS, the algorithm is used to monitor the patient's oxygenation index level in real-time, and the doctor can adjust the ventilator treatment plan at any time. On the other hand, the MIN–BER–FS algorithm can significantly reduce the amount

of computation, and it is easy to transplant the algorithm to the ordinary microprocessors of ventilators and monitors, which can realize a more intelligent aided diagnosis.

There were also some limitations associated with this study. The MIMIC–III database used in this study was a single-center database. Even though in the experimental design we separated the training set from the test set, we still need to perform external verification to ensure that the model works well in different hospitals. We found no data on bilateral pulmonary infiltrate of non-cardiogenic origin in the database, and there were very few patients with a clear diagnosis of ARDS. In the process of patient screening, this study based on the actual definition of the Berlin definition and database, as far as possible to select patients who meet the diagnostic criteria of ARDS, this process may introduce some confounding factors. The population of the clinical database we used was mainly concentrated at the age of 55, and most of the patients were middle-aged and elderly patients, which may bias our training model more focused on elderly patients, and there may be deviations in low-age populations.

Dataset imbalance is a ubiquitous problem in clinical research, and we have adopted a compromise of oversampling dataset imbalances. Oversampling does not solve the problem fundamentally, but can only alleviate the deviation of the results caused by data imbalances to some extent. The problem of data set imbalance can be fundamentally solved only by expanding the dataset. Missing data is an important problem in all modeling efforts, especially in the healthcare domain. For the MIMIC database, missing data problems also exist, such as the patient's noninvasive blood pressure, airway pressure, and other physiological parameters. If these missing data is omitted, a lot of samples will be lost. If some technical means are used for missing data processing, most of the samples will be retained. However, the above two methods will cause deviations in the model, the former was caused by the partial patient data loss, and the latter was caused by the introduction of some errors in the process of missing value estimation based on interpolation[19].

This study was exploratory and was mainly applied to investigate whether the use of noninvasive parameters could identify the ARDS patients, and to use feature selection techniques to select which noninvasive parameters yielded a higher correlation to the oxygenation level. In the future, we will include more patients with ARDS and develop multi-classification methods to achieve continuous ARDS disease severity identification. The outcomes of this study are expected to provide some ideas for future related research.

Next, in the future, an early warning system of the severity of ARDS for the monitors and ventilators will be developed using a multi-classification algorithm The patient's oxygenation level ($\leq$300 or >300) was identified by collecting mechanical ventilation parameters and physical signs to assist the physician in the diagnosis of ARDS without blood gas analysis. For patients who have been diagnosed as ARDS, the algorithm is used to monitor the patient's oxygenation index level in real-time, and the doctor can adjust the ventilator treatment plan at any time.

## 6. Conclusion

In conclusion, the overall classification effects of machine learning algorithms were better than those elicited by traditional algorithms. For machine learning algorithms, XGBoost was significantly better than the other three algorithms. Feature sorting and feature selection algorithms can help us understand the characteristics of ARDS to identify which features elicit better correlations, and can improve us design high-precision algorithms. The method can continually provide medical assistants with auxiliary diagnosis suggestions.

## Supporting information

**S1 Fig. The application scenario for the present algorithm.** The algorithm continuously monitors the patient's oxygenation level using basic patient information, ventilator

parameters, and monitoring parameters to help the doctor diagnose whether the patient has ARDS and adjust the treatment plan for ARDS patients.
(TIF)

**S1 Dataset. The train dataset.** The train dataset including 6601 patients (148,414 points).
(CSV)

**S2 Dataset. The test dataset.** The test dataset including 2101 patients (47,352 points).
(CSV)

## Author Contributions

**Conceptualization:** Guang Zhang.

**Formal analysis:** Guang Zhang.

**Methodology:** Taihu Wu, Ming Yu, Feng Chen, Jing Yuan.

**Project administration:** Taihu Wu, Ming Yu, Feng Chen.

**Resources:** Pengcheng Yang, Ming Yu.

**Software:** Pengcheng Yang, Chunchen Wang, Jiameng Xu.

**Supervision:** Jing Yuan, Guang Zhang.

**Validation:** Pengcheng Yang, Chunchen Wang, Jing Yuan, Jiameng Xu.

**Visualization:** Pengcheng Yang, Jing Yuan.

**Writing – original draft:** Pengcheng Yang.

**Writing – review & editing:** Pengcheng Yang, Guang Zhang.

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
