## [Decision Letter · Decision Letter 0]

24 Sep 2019

PONE-D-19-15059

A new method for identifying the severity of the acute respiratory distress syndrome disease based on noninvasive physiological parameters

PLOS ONE

Dear Mr Zhang,

Thank you for submitting your manuscript to PLOS ONE. After careful consideration, we feel that it has merit but does not fully meet PLOS ONE’s publication criteria as it currently stands. Therefore, we invite you to submit a revised version of the manuscript that addresses the points raised during the review process. For further details, please see the comments of the two expert reviewers below.

We would appreciate receiving your revised manuscript by Nov 08 2019 11:59PM. To enhance the reproducibility of your results, we recommend that if applicable you deposit your laboratory protocols in protocols.io, where a protocol can be assigned its own identifier (DOI) such that it can be cited independently in the future. For instructions see: http://journals.plos.org/plosone/s/submission-guidelines#loc-laboratory-protocols

We look forward to receiving your revised manuscript.

Kind regards,

Lars Kaderali

Academic Editor

PLOS ONE

Journal Requirements:

2. In ethics statement in the manuscript and in the online submission form, please provide additional information about the database used in your retrospective study. Specifically, please ensure that you have discussed whether all data were fully anonymized before you accessed them and/or whether the IRB or ethics committee waived the requirement for informed consent. If patients provided informed written consent to have their data used in research, please include this information.

Additional Editor Comments (if provided):

Reviewers' comments:

Reviewer's Responses to Questions

**Comments to the Author**

1. Is the manuscript technically sound, and do the data support the conclusions?

Reviewer #1: Partly

Reviewer #2: No

2. Has the statistical analysis been performed appropriately and rigorously? 

Reviewer #1: Yes

Reviewer #2: Yes

3. Have the authors made all data underlying the findings in their manuscript fully available?

Reviewer #1: Yes

Reviewer #2: Yes

4. Is the manuscript presented in an intelligible fashion and written in standard English?

Reviewer #1: Yes

Reviewer #2: Yes

5. Review Comments to the Author

Reviewer #1: This study uses the MIMIC III Database to test and validate an algorithm based on patient non-invasive physiological parameters to estimate P/F to aid in the diagnosis of ARDS disease. The authors chose parameters based on a literature review and chose : age, gender, height, weight, body mass index (BMI), ethnicity, ICU type, length of stay in ICU, admission type, in-hospital mortality, SpO2, temperature, heart rate, blood pressure, Glasgow Coma Scale (GCS), respiratory rate, tidal volume, minute ventilation, peak pressure, plateau pressure, mean air pressure, PEEP, FiO2, P/F, S/F, Oxygenation Index (OI), Oxygenation Saturation Index (OSI) to constructed a machine learning algorithm combined with the filter feature selection method to study the correlation of various noninvasive parameters from patients to identify ARDS. Cross-validation techniques are used to verify the performance of algorithms for different feature subsets. XGBoost using the optimal feature subset had the best performance of ARDS identification with the sensitivity of 84.03%, the specificity of 87.75% and the AUC of 0.9128. For the four machine learning algorithms, reducing a certain number of features AUC can still above 0.8. Compared to Rice Linear Model, this method has the advantages of high reliability and continually monitoring the development of patients with ARDS.

Questions:

1. As the authors chose patients with PaO2 /FIO2 <300 and > 300 for identification of ARDS it is not clear how they verified the severity of the disease and the hole and weight of the different parameters to characterize the ARDS severity as they stated. Please clarify…….

2. It is not clear how the clinicians can use this information to follow the ARDS patients in the clinical practice .Please clarify….

3. The authors concluded that XGBoost using the optimal feature subset had the best performance of ARDS identification with the sensitivity of 84.03%, the specificity of 87.75% and the AUC of 0.9128. Please discuss the explanation for this result and this result will add in the use of their algorithm in the future……

Reviewer #2: Introduction: The introduction is well written, contextualized very well the topic about the importance of early diagnosis and less invasive form of ARDS. It demonstrated the need for accurate algorithms with good sensitivity and specificity for early detection of ARDS and the levels of the P/F ratio without the need blood gas measured. It formulated the hypothesis of an algorithm that combines the analysis of vital signs, laboratory exams and clinical signs routinely recorded in the intensive care environment capable of predicting ARDS and its severity.

Materials and methods: The methodology of the study is well described, and the statistical tools used for measures the diagnostic accuracy of a test (sensitivity, specificity, positive predictive value, negative predictive value, accuracy and analysis of the ROC curve) was implemented.

I believe that it is necessary for the real and reliable diagnosis of ARDS to record data that proves hypoxemia and bilateral pulmonary infiltrate of non-cardiogenic origin (between lines 117 and 124 there is only the inclusion of data that proves hypoxemia and radiological alteration).

Results: In the analysis of the results it is important to present a comparative table between the diagnostic algorithm of best accuracy identified by the study (XGboost) and a Gold Standard test for the diagnosis of ARDS with the respective confidence intervals.

Conclusion: The conclusion is well written demonstrating the superiority of the XGboost algorithm in relation to the other algorithms studied, but it cannot be affirmed its full capacity in the diagnosis and graduation of ARDS for the reasons described previously.

6. PLOS authors have the option to publish the peer review history of their article (what does this mean?). If published, this will include your full peer review and any attached files.

Reviewer #1: Yes: Carmen Silvia Valente Barbas

Reviewer #2: Yes: Rodrigo Cruvinel Figueiredo

---

## [Author Response · Author response to Decision Letter 0]

9 Nov 2019

Reviewer1:

 As the authors chose patients with PaO2 /FIO2 <300 and > 300 for identification of ARDS, it is not clear how they verified the severity of the disease and the hole and weight of the different parameters to characterize the ARDS severity as they stated. Please clarify. 

Answer： 

Thank you for your thoughtful comment and good suggestion.

The reviewer1 questioned whether this study could verify the severity of ARDS disease. At present, the algorithm designed in this study has some limitations on the identification of the severity of ARDS disease. 

According to the 2012 Berlin definition, ARDS is characterized by the following:

1）lung injury of acute onset, within one week of an apparent clinical insult and with the progression of respiratory symptoms;

2）bilateral opacities on chest imaging;

3）respiratory failure not explained by heart failure or volume overload;

4）decreased PaO2/FiO2 ratio.

In the clinical practice, doctors need to combine these four points to diagnose the patient's condition. These indicators are mainly used for the diagnosis and classification of ARDS and are also used to predict the prognosis of patients with ARDS. In this study, we selected the P/F ratio as the primary indicator for diagnosing whether the patient's ARDS occurred.

Based on the above reasons, we designed the algorithm to establish the nonlinear relationship between a variety of non-invasive physiological parameters and the P/F ratio, for auxiliary diagnosis of ARDS disease according to the identified oxygenation level, reducing the frequency of blood gas analysis and save medical resources.

The importance of the features in this study is relative to the level of oxygenation index. As shown in Table 4 of the manuscript, these importance rankings reflect the feature ability to identify P/F > 300 or <= 300.

The Berlin definition divides ARDS disease into three levels: mild ARDS (200<P/F≤300), moderate ARDS (100<P/F≤200), severe ARDS (P/F≤100). Currently, we focused on patients with ARDS using an oxygenation index of 300 as the classification threshold. Therefore, this study can only identify patients with or without ARDS but cannot identify the ARDS severity of patients. In the future, we will include more patients with ARDS and develop multi-classification methods to achieve continuous ARDS disease severity identification.

This detailed description of the algorithm’s limitation of ARDS disease severity identification disability was added in the feature selection result and limitation section as reviewer1’ suggestion, as follows:

“Table 4 represents the normalized values of the scores provided by the three filter methods under consideration. The importance of the features in this study is relative to the level of oxygenation index.”

“This study was exploratory and was mainly applied to investigate whether the use of noninvasive parameters could identify the ARDS patients, and to use feature selection techniques to select which noninvasive parameters yielded a greater correlation to the oxygenation level. In the future, we will include more patients with ARDS and develop multi-classification methods to achieve continuous ARDS disease severity identification. The outcomes of this study are expected to provide some ideas for future related research.”

 It is not clear how the clinicians can use this information to follow the ARDS patients in the clinical practice. Please clarify.

Answer：

Thank you for your thoughtful comment.

Currently, the diagnosis and treatment of ARDS mainly depend on the oxygenation index. Therefore, the algorithm uses the non-invasive parameters of the patient to identify the current oxygenation level. Based on the identification results, the clinicians diagnose the patient's ARDS condition by comprehensively analyzing the patient's onset time, imaging examination, etc., thereby reducing the frequency of blood gas analysis. The algorithm can also continuously identify the patient's oxygenation level, and the doctor can adjust the ventilator treatment plan according to the identification result.

The non-invasive parameters involved in this algorithm are the patient's demographics, physiological parameters, respiratory parameters, etc. which can quickly obtain from ordinary monitors and ventilators. Later, a real-time P/F level monitoring system can be developed using the parameters involved in the optimal or minimum feature subset. For example, the XGBoost algorithm has the optimal feature subset, including the first 12 features, and the corresponding parameters are SpO2, FiO2, PEEP, Mean air pressure, Respiratory rate, Plateau pressure, TV, etc. Real-time acquisition of physiological parameters of the patient in the monitor and ventilator in the ICU and the oxygenation level of the patient at the current time is obtained by using this present algorithm, thereby assisting the doctor to complete the diagnosis of the ARDS without blood gas analysis.

ARDS is usually treated with mechanical ventilation and vital signal monitoring in the intensive care unit (ICU), we hope to integrate this algorithm into the ventilators or monitors to improve the intelligence of the ventilators and monitors. The ventilator and monitor can use the patient's multiple non-invasive physiological parameters to monitor the patient's oxygenation level in real-time without blood gas analysis, providing clinicians with diagnostic and therapeutic recommendations.

This detailed description on how to apply this algorithm was added in the discussion section as reviewer1’ suggestion, as follows:

 “Next, in the future, an early warning system of the severity of ARDS for the monitors and ventilators will be developed using a multi-classification algorithm The patient's oxygenation level (≤300 or >300) was identified by collecting mechanical ventilation parameters and physical signs to assist the physician in the diagnosis of ARDS without blood gas analysis. For patients who have been diagnosed as ARDS, the algorithm is used to monitor the patient's oxygenation index level in real-time, and the doctor can adjust the ventilator treatment plan at any time.”

 The authors concluded that XGBoost using the optimal feature subset had the best performance of ARDS identification with the sensitivity of 84.03%, the specificity of 87.75% and the AUC of 0.9128. Please discuss the explanation for this result and this result will add in the use of their algorithm in the future.

Answer： 

Thank you for your thoughtful comment.

Based on the feature ranking results, the MIN–BER–FS algorithm is used to find the optimal feature subset (minimum BER), and the minimum feature subset is located according to the BER standard deviation at the optimal feature subset. The smallest subset will discard some redundancy information, reduce the amount of data, save computing costs, and can be used in ordinary microprocessors for higher portability. In this study, we compare the experimental results of the minimum feature subset, the optimal feature subset, and the total feature subset, and can reflect the performance of the algorithm from multiple aspects.

From the results, XGBoost has the best results using the optimal subset. At the same time, XGBoost achieved better results than the other three algorithms in the minimum feature subset (only the first six features). Observing the variation of the standard deviation during the training process, we can see that AdaBoost has the best stability.

The application scenario for this algorithm is as shown in Figure 1. The patient's oxygenation level (≤300 or >300) was identified by collecting mechanical ventilation parameters and physical signs to assist the physician in the diagnosis of ARDS without blood gas analysis. For patients who have been diagnosed as ARDS, the algorithm is used to monitor the patient's oxygenation index level in real-time, and the doctor can adjust the ventilator treatment plan at any time. On the other hand, the MIN–BER–FS algorithm can significantly reduce the amount of computation, and it is easy to transplant the algorithm to the ordinary microprocessors of ventilators and monitors, which can realize a more intelligent aided diagnosis.

Fig 1. The application scenario for the present algorithm

This detailed description on the result and how to apply this algorithm was added in the results section as reviewer1’ suggestion, as follows:

“In this study, we analyze the classification ability of features, use filter methods to sort the importance of features, and use MIN–BER–FS algorithm to find the optimal feature subset and the minimum feature subset. In the algorithmic evaluation, we compare the experimental results of the minimum, optimal, and all feature subsets, which can reflect the ability of the algorithm to mine information from different aspects. In the process of feature traversal, it can also be said that the feature importance is consistent with the feature sorting experiment results. From the results, XGBoost has the best results under the optimal subset. At the same time, XGBoost achieved better results than the other three algorithms in the minimum feature subset (only the first six features). Observing the variation of the standard deviation during the training process, we can see that AdaBoost has the best stability.”

“At the same time, XGBoost achieved better results than the other three algorithms in the minimum feature subset (only the first six features). Observing the variation of the standard deviation during the training process, we can see that AdaBoost has the best stability.”

Reviewer2:

 Materials and methods: The methodology of the study is well described, and the statistical tools used for measures the diagnostic accuracy of a test (sensitivity, specificity, positive predictive value, negative predictive value, accuracy and analysis of the ROC curve) was implemented. I believe that it is necessary for the real and reliable diagnosis of ARDS to record data that proves hypoxemia and bilateral pulmonary infiltrate of non-cardiogenic origin (between lines 117 and 124 there is only the inclusion of data that proves hypoxemia and radiological alteration).

Answer：

Thank you for your thoughtful comment.

Through careful analysis of the MIMIC-III database, we found no data on the bilateral pulmonary infiltrate of non-cardiogenic origin in the database, and there is no precise diagnosis of ARDS. To ensure the availability of data, we have adequately studied and discussed it with the professional clinical team. In the process of patient selection, according to the Berlin definition and the actual situation of the database, patients who meet the diagnostic criteria of ARDS should be selected as much as possible. This process may introduce some confounding factors. In future research, we will add more patients with precise ARDS disease to further improve the algorithm identification.

This detailed description on missing the data on bilateral pulmonary infiltrate of non-cardiogenic origin in the database was added in the limitation section as reviewer2’ suggestion, as follows:

“We found no data on bilateral pulmonary infiltrate of non-cardiogenic origin in the database, and there were very few patients with a clear diagnosis of ARDS. In the process of patient screening, this study based on the actual definition of the Berlin definition and database, as far as possible to select patients who meet the diagnostic criteria of ARDS, this process may introduce some confounding factors.”

 Results: In the analysis of the results it is important to present a comparative table between the diagnostic algorithm of best accuracy identified by the study (XGBoost) and a Gold Standard test for the diagnosis of ARDS with the respective confidence intervals.

Answer：

Thank you for your thoughtful comment.

The gold standard for ARDS diagnosis - the Berlin definition includes: 

1）lung injury of acute onset, within one week of an apparent clinical insult and with the progression of respiratory symptoms;

2）bilateral opacities on chest imaging ;

3）respiratory failure not explained by heart failure or volume overload;

4）decreased PaO2/FiO2 ratio.

For the data available for ARDS patients recorded in the MIMIC-III database, the only quantitative indicator was the oxygenation index. At the same time, the oxygenation index is also the primary diagnostic and grading standard in ARDS. We chose the P/F value as the fundamental basis for the ARDS identification. Therefore, the oxygenation index (P/F ratio) determined by blood gas analysis was used as the gold standard. The method of identifying whether ARDS occurs by oxygenation level has been applied in previous studies [1,2]. For example, Rice et al. used S/F ratio to fit P/F ratio. This method has been used in the Hamilton ventilator to continuously monitor the oxygenation level utilizing the patient's S/F ratio. Therefore, in the algorithm results, we compare the experimental results of this study with Rice's preliminary findings.

In the future, we will add other ARDS diagnostic gold standards to improve the identification performance of the algorithm model by clinically obtaining patients with well-diagnosed ARDS.

1. Yehya N, Servaes S, Thomas NJ. Characterizing degree of lung injury in pediatric acute respiratory distress syndrome. Crit Care Med. 2015;43: 937–946. doi:10.1097/CCM.0000000000000867

2. Horhat FG, Gundogdu F, David LV, Boia ES, Pirtea L, Horhat R, et al. Early Evaluation and Monitoring of Critical Patients with Acute Respiratory Distress Syndrome (ARDS) Using Specific Genetic Polymorphisms. Biochem Genet. 2017;55: 204–211. doi:10.1007/s10528-016-9787-0

 Conclusion: The conclusion is well written demonstrating the superiority of the XGboost algorithm in relation to the other algorithms studied, but it cannot be affirmed its full capacity in the diagnosis and graduation of ARDS for the reasons described previously.

Answer:

Thank you for your thoughtful comment.

In the absence of blood gas analysis, we estimate the oxygenation level of the patient at the current time by collecting multiple non-invasive parameters of the patient. The clinician combines the patient's other medical information to make appropriate diagnostic and treatment decisions. According to the Berlin definition, the severity of ARDS disease was divided into three levels: mild ARDS (200<P/F<=300), moderate ARDS (100<P/F<=200), severe ARDS (P/F<=100). At present, we only developed an algorithm identifying whether the patient's ARDS occurs by evaluating the oxygenation level (using P/F value of 300). In the future, we will be developing a multi-classification study on the oxygenation index to achieve an early warning system for identifying the severity of ARDS disease.

This detailed description of the limitation in the diagnosis and graduation of ARDS was added in the discussion and conclusion sections as reviewer2’ suggestion, as follows:

“This study was exploratory and was mainly applied to investigate whether the use of noninvasive parameters could identify the ARDS patients, and to use feature selection techniques to select which noninvasive parameters yielded a greater correlation to the oxygenation level. In the future, we will include more patients with ARDS and develop multi-classification methods to achieve continuous ARDS disease severity identification. The outcomes of this study are expected to provide some ideas for future related research.”

“In conclusion, the overall classification effects of machine learning algorithms were better than those elicited by traditional algorithms. For machine learning algorithms, XGBoost was significantly better than the other three algorithms. Feature sorting and feature selection algorithms can help us understand the characteristics of ARDS to identify which features elicit better correlations, and can help us design high-precision algorithms. The method can continually provide medical assistants with auxiliary diagnosis suggestions.”

---

## [Decision Letter · Decision Letter 1]

11 Dec 2019

A new method for identifying the acute respiratory distress syndrome disease based on noninvasive physiological parameters

PONE-D-19-15059R1

Dear Dr. Zhang,

We are pleased to inform you that your manuscript has been judged scientifically suitable for publication and will be formally accepted for publication once it complies with all outstanding technical requirements.

With kind regards,

Lars Kaderali

Academic Editor

PLOS ONE

Additional Editor Comments (optional):

Reviewers' comments:

Reviewer's Responses to Questions

**Comments to the Author**

1. If the authors have adequately addressed your comments raised in a previous round of review and you feel that this manuscript is now acceptable for publication, you may indicate that here to bypass the “Comments to the Author” section, enter your conflict of interest statement in the “Confidential to Editor” section, and submit your "Accept" recommendation.

Reviewer #1: All comments have been addressed

Reviewer #2: All comments have been addressed

2. Is the manuscript technically sound, and do the data support the conclusions?

Reviewer #1: Yes

Reviewer #2: Yes

3. Has the statistical analysis been performed appropriately and rigorously? 

Reviewer #1: Yes

Reviewer #2: Yes

4. Have the authors made all data underlying the findings in their manuscript fully available?

Reviewer #1: Yes

Reviewer #2: Yes

5. Is the manuscript presented in an intelligible fashion and written in standard English?

Reviewer #1: Yes

Reviewer #2: Yes

6. Review Comments to the Author

Reviewer #1: The authors adequately answered the reviewer questions and made the necessary changes in the revised version of the manuscript.

Reviewer #2: Doubts were explained by the author and the changes recorded in the original text. Congratulations!!!

7. PLOS authors have the option to publish the peer review history of their article (what does this mean?). If published, this will include your full peer review and any attached files.

Reviewer #1: Yes: Carmen Silvia Valente Barbas

Reviewer #2: Yes: Figueiredo, R. C.

---

## [Editor Report · Acceptance letter]

22 Jan 2020

PONE-D-19-15059R1 

A new method for identifying the acute respiratory distress syndrome disease based on noninvasive physiological parameters 

Dear Dr. Zhang:

I am pleased to inform you that your manuscript has been deemed suitable for publication in PLOS ONE. Congratulations! Your manuscript is now with our production department. 

With kind regards,

on behalf of

Prof. Dr. Lars Kaderali 

Academic Editor

PLOS ONE